# Analytically Tractable Bayesian Deep Q-Learning

## Abstract

Reinforcement learning (RL) has gained increasing interest since the demonstration
it was able to reach human performance on video game benchmarks using *deep
Q-learning* (DQN). The current consensus for training neural networks on such
complex environments is to rely on gradient-based optimization. Although alterna-
tive Bayesian deep learning methods exist, most of them still rely on gradient-based
optimization, and they typically do not scale on benchmarks such as the Atari game
environment. Moreover none of these approaches allow performing the analytical
inference for the weights and biases defining the neural network. In this paper, we
present how we can adapt the temporal difference Q-learning framework to make
it compatible with the *tractable approximate Gaussian inference* (TAGI), which
allows learning the parameters of a neural network using a closed-form analytical
method. Throughout the experiments with on- and off-policy reinforcement learn-
ing approaches, we demonstrate that TAGI can reach a performance comparable to
backpropagation-trained networks while using fewer hyperparameters, and without
relying on gradient-based optimization.

## 1   Introduction

Reinforcement learning (RL) has gained increasing interest since the demonstration it was able to
reach human performance on video game benchmarks using *deep Q-learning* (DQN) [17, 26]. Deep
RL methods typically require an explicit definition of an exploration-exploitation function in order to
compromise between using the current policy and exploring the potential of new actions. Such an
issue can be mitigated by opting for a Bayesian approach where the selection of the optimal action to
follow is based on Thompson sampling [23]. Bayesian deep learning methods based on variational
inference [12, 10, 5, 14, 20, 29], Monte-Carlo dropout [8], or Hamiltonian Monte-Carlo sampling
[18] have shown to perform well on regression and classification benchmarks, despite being generally
computationally more demanding than their deterministic counterparts. Note that none of these
approaches allow performing the analytical inference for the weights and biases defining the neural
network. Goulet et al. [9] recently proposed the *tractable approximate Gaussian inference* (TAGI)
method which allows learning the parameters of a neural network using a closed-form analytical
method. For convolutional architectures applied on classification benchmarks, this approach was
shown to exceed the performance of other Bayesian and deterministic approaches based on gradient
backpropagation, and to do so while requiring a smaller number of training epochs [19].

In this paper, we present how can we adapt the temporal difference Q-learning framework [24, 28] to
make it compatible with TAGI. Section 2 first reviews the theory behind TAGI and the expected value
formulation through the Bellman's Equation. Then, we present how the action-value function can
be learned using TAGI. Section 3 presents the related work associated with Bayesian reinforcement
learning, and Section 4 compares the performance of a simple TAGI-DQN architecture with the one
obtained for its backpropagation-trained counterpart.

## 2 TAGI-DQN Formulation

This section presents how to adapt the DQN frameworks in order to make them compatible with analytical inference. First, Section 2.1 reviews the fundamental theory behind TAGI, and Section 2.1 reviews the concept of long-term expected value through the Bellman's equation [25]. Then, Section 2.3 presents how to make the Q-learning formulation [28] compatible with TAGI.

### 2.1 Tractable Approximate Gaussian Inference

TAGI [9] relies on two main steps; *forward uncertainty propagation* and *backward update*. The first forward uncertainty propagation step is intended to build the joint prior between the neural network parameters and the hidden states. This operation is made by propagating the uncertainty from the model parameters and the input layer through the neural network. TAGI relies on the Gaussian assumption for the prior of parameters as well as for the variables in the input layer. In order to maintain the analytical tractability of the forward step, we rely on the *Gaussian multiplicative approximation* (GMA) which consists in approximating the product of two Gaussians by a Gaussian random variable whose moments match those calculated exactly using moment generating functions. In order to propagate uncertainty through non-linear activation functions, a second approximation made by locally linearizing these function at the expected value of the hidden unit being activated. Although this linearization procedure may seems to be a crude approximation, it has been shown to match or exceeds the state-of-the-art performance on fully-connected neural networks (FNN) [9], as well as convolutional neural networks (CNN) and generative adversarial networks [19]. TAGI succeeds in maintaining a linear computational complexity for the forward steps, (1) by assuming a diagonal covariance for all parameters in the network and for all the hidden units within a same layer, and (2) by adopting a layer-wise approach where the joint prior is only computed and stored for the hidden units on pairs of successive hidden layers, as well as the hidden units within a layer and the parameters connecting into it. This layer-wise approach is allowed by the inherent conditional independence that is built-in feed-forward neural network architectures.

The second backward update-step consists in performing layer-wise recursive Bayesian inference which goes from hidden-layer to hidden-layer and from hidden-layer to the parameters connecting into it. Given the Gaussian approximation for the joint prior throughout the network, the inference can be done analytically while still maintaining a linear computational complexity with respect to the number of weight parameters in the network. TAGI allows inferring the diagonal posterior knowledge for weights and bias parameters, either using one observation at a time, or using mini-batches of data. As we will show in the next sections, this online learning capacity is best suited for RL problems where we experience episodes sequentially and where we need to define a tradeoff between exploration and exploitation, as a function of our knowledge of the expected value associated with being in a state and taking an action.

### 2.2 Expected Value and Bellman's Equation

We define $r(\boldsymbol{s}, a, \boldsymbol{s}')$ as the reward for being in a state $\boldsymbol{s} \in \mathbb{R}^{\mathsf{S}}$, taking an action $a \in \mathcal{A} = \{a_1, a_2, \cdots a_{\mathsf{A}}\}$, and ending in a state $\boldsymbol{s}' \in \mathbb{R}^{\mathsf{S}}$. For simplicity, we use the short-form notation for the reward $r(\boldsymbol{s}, a, \boldsymbol{s}') \equiv r(\boldsymbol{s})$ in order to define the value as the infinite sum of discounted rewards

$$v(\boldsymbol{s}) = \sum_{k=0}^{\infty} \gamma^k r(\boldsymbol{s}_{t+k}). \tag{1}$$

As we do not know what will be the future states $\boldsymbol{s}_{t+k}$ for $k > 0$, we need to consider them as random variables ($\boldsymbol{S}_{t+k}$), so that the value $V(\boldsymbol{s}_t)$ becomes a random variable as well,

$$V(\boldsymbol{s}_t) = r(\boldsymbol{s}_t) + \sum_{k=1}^{\infty} \gamma^k r(\boldsymbol{S}_{t+k}). \tag{2}$$

Rational decisions regarding which action to take among the set $\mathcal{A}$ is based the maximization of the expected value as defined by the *action-value* function

$$q(\boldsymbol{s}_t, a_t) = \mu_V \equiv \mathbb{E}[V(\boldsymbol{s}_t, a_t, \pi)] = r(\boldsymbol{s}_t) + \mathbb{E}\left[\sum_{k=1}^{\infty} \gamma^k r(\boldsymbol{S}_{t+k})\right], \tag{3}$$

where it is assumed that at each time $t$, the agent takes the action defined in the policy $\pi$. In the case of episode-based learning where the agent interacts with the environment, we assume we know the tuple of states $\boldsymbol{s}_t$ and $\boldsymbol{s}_{t+1}$, so that we can redefine the value as

$$
\begin{aligned}
V(\boldsymbol{s}_t, a_t) &= r(\boldsymbol{s}_t) + \gamma \left( r(\boldsymbol{s}_{t+1}) + \sum_{k=1}^{\infty} \gamma^k r(\boldsymbol{S}_{t+1+k}) \right) \\
&= r(\boldsymbol{s}_t) + \gamma V(\boldsymbol{s}_{t+1}, a_{t+1}).
\end{aligned} \tag{4}
$$

Assuming that the value $V \sim \mathcal{N}(v; \mu_V, \sigma_V^2)$ in Equations 2 and 4 is described by Gaussian random variables, we can reparameterize these equations as the sum of the expected value $q(\boldsymbol{s}, a)$ and a zero-mean Gaussian random variable $\mathcal{E} \sim \mathcal{N}(\epsilon; 0, 1)$, so that

$$
V(\boldsymbol{s}, a) = q(\boldsymbol{s}, a) + \sigma_V \mathcal{E}, \tag{5}
$$

where the variance $\sigma_V^2$ and $\mathcal{E}$ are assumed here to be independent of $\boldsymbol{s}$ and $a$. Although in a more general framework this assumption could be relaxed, such an heteroscedastic variance term is outside from the scope of this paper. Using this reparameterization, we can write Equation 4 as the discounted difference between the expected values of two subsequent states

$$
\begin{aligned}
q(\boldsymbol{s}_t, a_t) &= r(\boldsymbol{s}_t) + \gamma q(\boldsymbol{s}_{t+1}, a_{t+1}) - \sigma_{V_t} \mathcal{E}_t + \gamma \sigma_{V_{t+1}} \mathcal{E}_{t+1} \\
&= r(\boldsymbol{s}_t) + \gamma q(\boldsymbol{s}_{t+1}, a_{t+1}) + \sigma_V \mathcal{E}.
\end{aligned} \tag{6}
$$

Note that in Equation 6, $\sigma_{V_t}$ and $\gamma \sigma_{V_{t+1}}$ can be combined in a single standard deviation parameters $\sigma_V$ with the assumption that $\mathcal{E}_i \perp\!\!\!\perp \mathcal{E}_j, \forall i \neq j$.

In the case where at a time $t$, we want to update the Q-values encoded in the neural net only after observing $n$-step returns [15], we can reformulate the observation equation so that

$$
q(\boldsymbol{s}_t, a_t) = \sum_{i=0}^{n-t-1} \gamma^i r(\boldsymbol{s}_{t+i}) + \gamma^{n-t} q(\boldsymbol{s}_n, a_n) + \sigma_V \mathcal{E}_t, \forall t = \{1, 2, \cdots, n-1\}. \tag{7}
$$

Note that in the application of Equation 7, we employ the simplifying assumption that $\mathcal{E}_t \perp\!\!\!\perp \mathcal{E}_{t+i}, \forall i \neq 0$, as Equation 6 already makes simplifying assumptions for the independence of $\sigma_V^2$ and $\mathcal{E}$. Note that in a more general framework, this assumption could be relaxed. An example of $n$-step returns is presented in the the algorithm displayed in §1 from the supplementary material.

The following subsections will present, for the case of categorical actions, how to model the deterministic action-value function $q(\boldsymbol{s}, a)$ using a neural network.

## 2.3 TAGI Deep Q-learning for Categorical Actions

Suppose we represent the environment's state at a time $t$ and $t + 1$ by $\{\boldsymbol{s}, \boldsymbol{s}'\}$, and the expected value for each of the A possible actions $a \in \mathcal{A}$ by the vector $\boldsymbol{q} \in \mathbb{R}^{\text{A}}$. In that context, the role of the neural network is to model the relationships between $\{\boldsymbol{s}, a\}$ and $\boldsymbol{q}$. Figure 1a presents a directed acyclic graph (DAG) describing the interconnectivity in such a neural network, where red nodes denote state variables, green nodes are vectors of hidden units $\boldsymbol{z}$, the blue box is a compact representation for the structure of a convolutional neural network, and where gray arrows represent the weights and bias $\boldsymbol{\theta}$ connecting the different hidden layers. Note that unlike other gray arrows, the red ones in (b) are not directed arcs representing dependencies, but they simply outline the flow of information that takes place during the inference step. For simplification purposes, the convolutional operations are omitted and all regrouped under the CNN box [19]. In order to learn the parameters $\boldsymbol{\theta}$ of such a network, we need to expand the graph from Figure 1a to include the reward $r$, the error term $\sigma_V \epsilon$, and $\boldsymbol{q}'$, the q-values of the time step $t + 1$. This configuration is presented in Figure 1b where the nodes that have been doubled represent the states $\boldsymbol{s}$ and $\boldsymbol{s}'$ which are both evaluated in a network sharing the same parameters. When applying Equation 6, q-values corresponding to a specific action can be selected using a vector $\boldsymbol{h}_i \in \{0, 1\}^{\text{A}}$ having a single non-zero value for the $i$-th component identifying which action was taken at a time $t$ so that

$$
q_i = [\boldsymbol{q}]_i = \boldsymbol{h}_i^{\mathsf{T}} \boldsymbol{q}. \tag{8}
$$

During the network's training, analogously to Thompson sampling [23], the vector $\boldsymbol{h}'_i \in \{0, 1\}^{\text{A}}$ is defined such that the $i$-th non-zero value corresponds to the index of the largest value among $\boldsymbol{q}'$, a

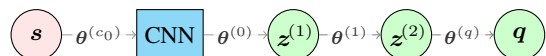

(a) Neural network DAG for modelling the action-value function $q$

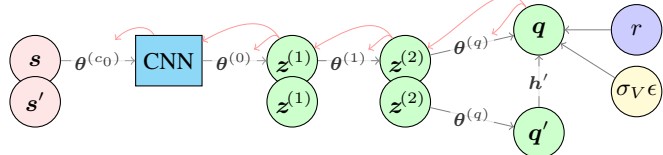

(b) DAG for the temporal-difference Q-learning configuration

Figure 1: Graphical representation of a neural network structure for temporal-difference Q-learning with categorical actions. The red nodes denote state variables, green nodes are vectors of hidden units $z$, and the blue box is a compact representation for the structure of a convolutional neural network. The gray arrows represent the weights and bias $\theta$ connecting the different hidden layers and the red arrows outline the flow of information that takes place during the inference step.

vector of realizations from the neural network's posterior predictive output $Q \sim \mathcal{N}(q'; \mu_{Q|\mathcal{D}}, \Sigma_{Q|\mathcal{D}})$. Because of the Gaussian assumptions in TAGI, this posterior predictive is readily available from the forward uncertainty propagation step, as outlined in §2.1.

The red arrows in Figure 1b outline the flow of information during the inference procedure. The first step consists in inferring $q$ using the relationships defined in either Equation 6 or 7. As this is a linear equation involving Gaussian random variables, the inference is analytically tractable. From there, one can follow the same layer-wise recursive procedure proposed by Goulet et al. [9] in order to learn the weights and biases in $\theta$. With the exclusion of the standard hyperparameters related to network architecture, batch size, buffer size or the discount factor, this TAGI-DQN framework only involves a single hyperparameter, $\sigma_V$, the standard deviation for the value function. Note that when using CNNs with TAGI, Nguyen and Goulet [19] recommended using a decay function for the standard deviation of the observation noise so that at after seing $e$ batches of $n$-steps,

$$\sigma_V^e = \max(\sigma_V^{\min}, \eta \cdot \sigma_V)^{e-1}. \tag{9}$$

The model in Equation 9 has three hyperparameters, the minimal noise parameter $\sigma_V^{\min}$, the decay factor $\eta$ and the initial noise parameter $\sigma_V$. As it was shown by Nguyen and Goulet [19] for CNNs and how we show in §4 for RL problems, TAGI's performance is robust towards the selection of these hyperparameters.

A comparison of implementation between TAGI and backpropagation on deep Q-network with experience replay [17] is shown in Figure 2. A practical implementation of $n$-step TAGI deep Q-learning is presented in Algorithm 1 from the supplementary material.

## 3 Related Works

Over the last decades, several approximate methods have been proposed in order to allow for Bayesian neural networks [18, 12, 10, 5, 14, 20, 29, 8] with various degree of approximations. Although some these methods have shown to be capable of tackling classification tasks on datasets such ImageNet [20], few of them have been applied on large-scale RL benchmark problems. The key idea behind using Bayesian methods for reinforcement learning is to consider the uncertainty associated with Q-functions in order to identify a tradeoff between exploring the performance of possible actions and exploiting the current optimal policy [25]. This typically takes the form of performing Thompson sampling [23] rather than relying on heuristics such as $\epsilon$-greedy.

For instance, MC dropout [8] was introduced has a method intrinsically suited for reinforcement learning. Nevertheless, five years after its inception, the approach has not yet been reliably scaled to more advanced benchmarks such as the Atari game environment. The same applies to Bayes-by-backprop [5] which was recently applied to simple RL problems [13], and which has not yet been applied to more challenging environments requiring convolutional networks. On the other hand, Bayesian neural networks relying on sampling methods such as Hamiltonian Monte-Carlo

| **Algorithm 1:** TAGI-DQN with Experience Replay | **Algorithm 2:** DQN with Experience Replay |
|---|---|
| 1 Initialize replay memory $\mathcal{R}$ to capacity $N$; $\mathbf{\Sigma}_V$; | 1 Initialize replay memory $\mathcal{R}$ to capacity $N$; |
| 2 Initialize parameters $\boldsymbol{\theta}$; | 2 Initialize parameters $\boldsymbol{\theta}$; |
| 3 Discount factor $\gamma$; | 3 Discount factor $\gamma$; |
| 4 **for** *episode* $= 1 : \mathbb{E}$ **do** | 4 Define $\epsilon$ (epsilon-greedy function); |
| 5     Reset environment $\mathbf{s}_0$; | 5 **for** *episode* $= 1 : \mathbb{E}$ **do** |
| 6     **for** $t = 1 : \mathbb{T}$ **do** | 6     Reset environment $\mathbf{s}_0$; |
| 7         $q(s_t, a) : Q(s_t, a) \sim \mathcal{N}(\boldsymbol{\mu}_{\boldsymbol{\theta}}^Q(s_t, a), \mathbf{\Sigma}_{\boldsymbol{\theta}}^Q(s_t, a))$; | 7     **for** $t = 1 : \mathbb{T}$ **do** |
| 8         $a_t = \arg\max\limits_{a \in \mathcal{A}} q(s_t, a)$; | 8         $u : U \sim \mathcal{U}(0, 1)$; |
| 9         $s_{t+1}, r_t = $ enviroment$(a_t)$; | 9         $a_t = \begin{cases} \texttt{randi(A)} & u < \epsilon; \\ \arg\max\limits_{a \in \mathcal{A}} Q_{\boldsymbol{\theta}}(s_t, a) & u \geq \epsilon; \end{cases}$ |
| 10         Store $\{s_t, a_t, r_t, s_{t+1}\}$ in $\mathcal{R}$; | 10         $s_{t+1}, r_t = $ enviroment$(a_t)$; |
| 11         Sample random batch of $\{s_j, a_j, r_j, s_{j+1}\}$; | 11         Store $\{s_t, a_t, r_t, s_{t+1}\}$ in $\mathcal{R}$; |
| 12         $q(s_{j+1}, a') : Q(s_{j+1}, a') \sim \mathcal{N}(\boldsymbol{\mu}_{\boldsymbol{\theta}}^Q(s_{j+1}, a'), \mathbf{\Sigma}_{\boldsymbol{\theta}}^Q(s_{j+1}, a'))$; | 12         Sample random batch of $\{s_j, a_j, r_j, s_{j+1}\}$; |
| 13         $a'_{j+1} = \arg\max\limits_{a' \in \mathcal{A}} q(s_{j+1}, a')$; | 13         $y_j = r_j + \gamma \max\limits_{a' \in \mathcal{A}} Q_{\boldsymbol{\theta}}(s_{j+1}, a')$; |
| 14         $\boldsymbol{\mu}_j^y = r_j + \gamma \boldsymbol{\mu}_{\boldsymbol{\theta}}^Q(s_{j+1}, a'_{j+1})$; | 14         Update $\boldsymbol{\theta}$ using gradient descent on |
| 15         $\mathbf{\Sigma}_j^y = \gamma^2 \mathbf{\Sigma}_{\boldsymbol{\theta}}^Q(s_{j+1}, a'_{j+1}) + \mathbf{\Sigma}_V$; | 15         $L = 0.5\left[y_j - Q_{\boldsymbol{\theta}}(s_j, a_j)\right]^2$; |
| 16         Update $\boldsymbol{\theta}$ using TAGI on PDF$(\boldsymbol{\theta}|\mathbf{y})$ | |

Figure 2: Comparison of TAGI with backpropagation on deep Q-network with experience replay. PDF: probability density function; $L$: loss function; $\mathcal{U}$: uniform distribution; `randi`: uniformly distributed pseudorandom integers.

[18] are typically computationally demanding to be scaled to RL problems involving such a complex environment.

Although mainstream methods related to Bayesian neural networks have seldom been applied to complex RL problems, several research teams have worked on alternative approaches in order to allow performing Thompson sampling. For instance, Azizzadenesheli et al. [4] have employed a deep Q-network where the output layer relies on Bayesian linear regression. This approach was shown to be outperforming its deterministic counterparts on Atari games. Another approach by Osband et al. [21] employs bootstrapped deep Q-networks with multiple network heads in order to represent the uncertainty in the Q-functions. This approach was also shown to scale to Atari games while presenting an improved performance in comparison with deterministic deep Q-networks. Finally, Wang and Zhou [27] have tackled the same problem, but this time by modelling the variability in the Q-functions through a latent space learned using variational inference. Despite its good performance on the benchmarks tested, it did not allowed to be scaled to the Atari game environment.

The TAGI deep Q-network presented in th is paper is the first demonstration that an analytically tractable inference approach for Bayesian neural networks can be scaled to a problem as challenging as the Atari game environment.

## 4   Benchmarks

This section compares the performance of TAGI with backpropagation-based standard implementations on off- and on-policy deep RL. For the off-policy RL, both TAGI-based and backpropagation-based RL approaches are applied to deep Q-learning with experience replay (see Algorithm 1 & 2) for the lunar lander and cart pole environments. For the on-policy RL, TAGI is applied to the $n$-step Q-learning algorithm and is compared with its backpropagation-based counterpart [15]. We perform the comparison for five Atari games including Beamrider, Breakout, Pong, Qbert, and Space Invaders. Note that these five games are commonly selected for tuning hyperparameters for the entire Atari games [15, 16]. All benchmark environments are taken from the OpenAI Gym [6].

## 4.1 Experimental Setup

In the first experiments with off-policy RL, we use a fully-connected multilayer perceptron (MLP) with two hidden layers of 256 units for the lunar lander environment, and with one hidden layer of 64 units for the cart pole environment. In these experiments, there is no need for input processing nor for reward normalization. Note that unlike for the deterministic Q-network, TAGI does not use a target Q-network for ensuring the stability during training and allows eliminating the hyperparameter related to the target update frequency. For the deep Q-network trained with backpropagation, we employ the pre-tuned implementation of OpenAI baselines [7] with all hyperparameters set to the default values.

For the Atari experiments with on-policy RL, we use the same input processing and model architecture as Mnih et al. [15]. The Q-network uses two convolutional layers (16-32) and a full-connected MLP of 256 units. TAGI $n$-step Q-learning only uses a single network to represent the value function for each action, and relies on a single learning agent. The reason behind this choice is that TAGI current main library is only available on Matlab which does not support running a Python multiprocessing module such as the OpenAI gym. In the context of TAGI, we use an horizon of 128 steps and as recommended by Andrychowicz et al. [3] and following practical implementation details [1, 2], each return in $n$-step Q-learning algorithm is normalized by subtracting the average return from the current $n$-steps and then dividing by the empirical standard deviation from the set of $n$ returns. The standard deviation for the value function, $(\sigma_V)$, is initialized at 2. $\sigma_V$ is decayed each 128 steps with a factor $\eta = 0.9999$. The minimal standard deviation for the value function $\sigma_V^{\min} = 0.3$. These hyperparameters values were not grid-searched but simply adapted to the scale of the problems and are kept constant for all experiments. The complete details of the network architecture and hyperparameters are provided in the supplementary material.

## 4.2 Results

For the first set of experiments using off-policy RL, Figure 3 presents the average reward over 100 episodes for three runs for the lunar lander and cart pole environment. The TAGI-based deep Q-learning with experience replay shows a faster and more stable learning than the one relying on backpropagation, while not requiring a target network.

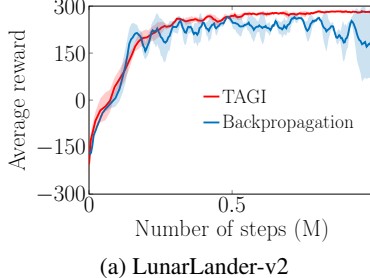
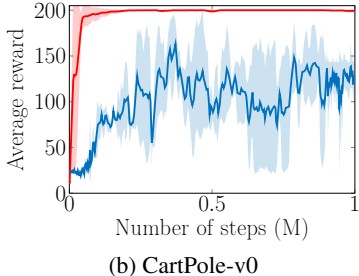

(a) LunarLander-v2          (b) CartPole-v0

Figure 3: Illustration of average rewards over 100 episodes of three runs for one million time steps for the TAGI-based and backpropagation-based deep Q-learning.

Table 1 shows that the average reward over the last 100 episodes obtained using TAGI are greater than the one obtained using backpropagation.

Table 1: Average reward over the last 100 episodes for the lunar lander and cart pole experiments. TAGI: Tractable Approximate Gaussian Inference.

| Method | Lunar lander | Cart pole |
|---|---|---|
| TAGI | $277.6 \pm 6.3$ | $199.2 \pm 1.3$ |
| Backpropagation | $166.7 \pm 103.6$ | $130.3 \pm 16.9$ |

Figure 4 compares the average reward over 100 episodes for three runs obtained for TAGI, with the results from Mnih et al. [15] for the second set of experiments on Atari games. Note that all results presented were obtained for a single agent, and that the results for the backpropagation-trained networks are only reported at the end of each epoch.

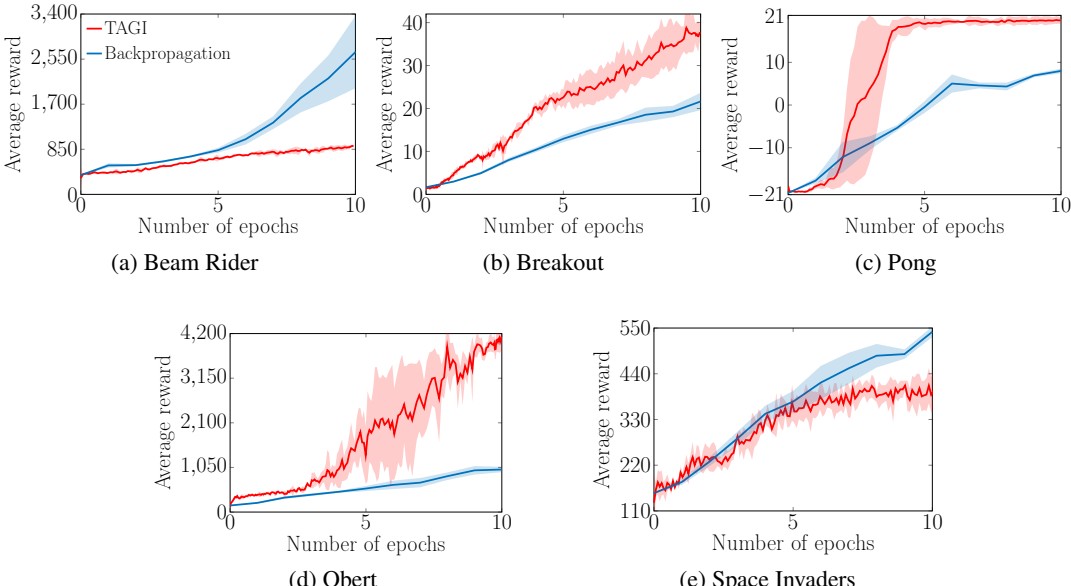

(a) Beam Rider        (b) Breakout        (c) Pong

(d) Qbert        (e) Space Invaders

Figure 4: Illustration of average reward over 100 episodes of three runs for five Atari games. The number of epochs is used here for the comparison of TAGI and backpropagation-trained counterpart obtained by Mnih et al. [15]. Each epoch corresponds to four million frames. The environment identity are {*Atari Game*}NoFrameSkip-v4.

Results show that TAGI outperforms the results from the original $n$-step Q-learning algorithm trained with backpropagation [15] on Breakout, Pong, and Qbert, while underperforming on Beam Rider and Space Invaders. The average training time of TAGI for an Atari game is approximately 13 hours on GPU calculations benchmarked on a 4-core-intel desktop of 32 GB of RAM with a NVIDIA GTX 1080 Ti GPU. The training speed of TAGI for the experiment of the off-policy deep RL is approximately three times slower on CPU calculations than the backpropagation-trained counterpart. The reason behind this slower training time is because of its intrinsically different inference engine, so that TAGI's implementation is not compatible with existing libraries such as TensorFlow or Pytorch. TAGI's library development is still ongoing and it is not yet fully optimized for computational efficiency. Overall, these results for on- and off policy RL approaches confirm that TAGI can be applied to large scale problems such as deep Q-learning.

## 5   Discussion

Although the performance of TAGI does not systematically outperform its backpropagation-based counterpart, it requires fewer hyperparameters (see §3 in supplementary material). This advantage is one of the key aspects for improving the generalization and reducing the computational cost of the hyperparameter tuning process which are the key challenges in current state of deep RL [11]. For instance, in this paper, the TAGI's hyperparameters relating to the standard deviation of value function ($\sigma_V$) are kept constant across all experiments. Moreover, since these hyperparameters were not subject to grid-search in order to optimize the performance, the results obtained here are representative of what a user should obtain by simply adapting the hyperparameters to fit the specificities and scale of the environment at hand.

More advanced RL approaches such as advanced actor critic (A2C) [15] and proximal policy optimization (PPO) [22] employ two-networks architectures in which one network is used to approximate a value function and other is employed to encode the policy. The current TAGI-RL framework is not yet able to handle such architectures because training a policy network involves an optimization problem for the selection of the optimal action. Backpropagation-based approach currently rely on gradient optimization to perform this task, while TAGI will require developing alternative approaches in order to maintain the analytical tractability without relying on gradient-based optimization.

## 6 Conclusion

This paper presents how to adapt TAGI to deep Q-learning; Throughout the experiments, we demonstrated that TAGI could reach a performance comparable to backpropagation-trained networks while using fewer hyperparameters. These results challenge the common belief that for large scale problems such as the Atari environment, neural networks can only be trained by relying on gradient backpropagation. We have shown here that this current paradigm is no longer the only alternative as TAGI has a linear computational complexity and can be used to learn the parameters complex networks in an analytically tractable manner, without relying on gradient-based optimization.

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
