# Supplementary Material: Analytically Tractable Bayesian Deep Q-Learning

## 1 Algorithm

This section presents the $n$-steps Q-learning algorithm with Tractable Approximate Gaussian Inference (TAGI).

---
**Algorithm 1:** $n$-step Q-learning with TAGI

---
**1** Initialize $\boldsymbol{\theta}$ ; $\boldsymbol{\Sigma}_V$; number of steps $(N)$
**2** Initialize memory $\mathcal{R}$ to capacity $N$;
**3** steps = 0;
**4** **for** *episode* $= 1 :$ E **do**
**5**      Reset environment $\mathbf{s}_0$;
**6**      **for** $t = 1 :$ T **do**
**7**          steps = steps + 1;
**8**          $q(s_t, a) : Q(s_t, a) \sim \mathcal{N}(\boldsymbol{\mu}_{\boldsymbol{\theta}}^Q(s_t, a), \boldsymbol{\Sigma}_{\boldsymbol{\theta}}^Q(s_t, a))$;
**9**          $a_t = \underset{a \in \mathcal{A}}{\arg\max} \, q(s_t, a)$;
**10**          $s_{t+1}, r_t = $ enviroment$(a_t)$;
**11**          Store $\{s_t, a_t, r_r\}$ in $\mathcal{R}$;
**12**          **if** *steps mod* $N == 0$ **then**
**13**             $q(s_{t+1}, a') : Q(s_{t+1}, a') \sim \mathcal{N}(\boldsymbol{\mu}_{\boldsymbol{\theta}}^Q(s_{t+1}, a'), \boldsymbol{\Sigma}_{\boldsymbol{\theta}}^Q(s_{t+1}, a'))$;
**14**             $a'_{t+1} = \underset{a \in \mathcal{A}}{\arg\max} \, q(s_{t+1}, a')$;
**15**             Take $N$ samples of $\{s_j, a_j, r_j\}$ from $\mathcal{R}$;
**16**             $\mu_N^y = \boldsymbol{\mu}_{\boldsymbol{\theta}}^Q(s_{t+1}, a'_{t+1}); \Sigma_N^y = \boldsymbol{\Sigma}_{\boldsymbol{\theta}}^Q(s_{t+1}, a'_{t+1})$;
**17**             **for** $j = N - 1 : 1$ **do**
**18**                $\mu_j^y = r_j + \gamma \mu_{j+1}^y; \Sigma_j^y = \gamma^2 \Sigma_{j+1}^y + \Sigma_V$;
**19**             Update $\boldsymbol{\theta}$ using TAGI;
**20**             Initialize memory $\mathcal{R}$ to capacity $N$;

---

## 2 Model Architecture

This appendix contains the specifications for each model architecture in the experiment section. $D$ refers to a layer depth; $W$ refers to a layer width; $H$ refers to the layer height in case of convolutional or pooling layers; $K$ refers to the kernel size; $P$ refers to the convolutional kernel padding; $S$ refers

Submitted to 35th Conference on Neural Information Processing Systems (NeurIPS 2021). Do not distribute.

9   to the convolution stride; $\sigma$ refers to the activation function type; ReLU refers to rectified linear unit; $N_a$ refers to the number of actions.

Table 1: Model Architecture for Cart pole

| Layer | $D \times W \times H$ | $K \times K$ | $P$ | S | $\sigma$ |
|---|---|---|---|---|---|
| Input | $4 \times 1 \times 1$ | - | - | - | - |
| Full connected | $64 \times 1 \times 1$ | - | - | - | ReLU |
| Output | $2 \times 1 \times 1$ | - | - | - | - |

Table 2: Model Architecture for Lunar lander

| Layer | $D \times W \times H$ | $K \times K$ | $P$ | S | $\sigma$ |
|---|---|---|---|---|---|
| Input | $8 \times 1 \times 1$ | - | - | - | - |
| Full connected | $256 \times 1 \times 1$ | - | - | - | ReLU |
| Full connected | $256 \times 1 \times 1$ | - | - | - | ReLU |
| Output | $4 \times 1 \times 1$ | - | - | - | - |

Table 3: Model Architecture for Atari domain

| Layer | $D \times W \times H$ | $K \times K$ | $P$ | S | $\sigma$ |
|---|---|---|---|---|---|
| Input | $4 \times 84 \times 84$ | - | - | - | - |
| Convolutional | $16 \times 20 \times 20$ | $8 \times 8$ | 0 | 4 | ReLU |
| Convolutional | $32 \times 9 \times 9$ | $4 \times 4$ | 0 | 2 | ReLU |
| Full connected | $256 \times 1 \times 1$ | - | - | - | ReLU |
| Output | $N_a \times 1 \times 1$ | - | - | - | |

10

## 3   Hyperparameter

12   This appendix details the hyperparameters for each model architecture in the experiment section

Table 4: Hyperparameters for Cart pole and Lunar lander

| Method | # | Hyperparameter | Value |
|---|---|---|---|
| TAGI | 1 | Initial standard deviation for the value function ($\sigma_V$) | 2 |
| | 2 | Decay factor ($\eta$) | 0.9999 |
| | 3 | Minimal standard deviation for the value function ($\sigma_V^{\min}$) | 0.3 |
| | 4 | Buffer size | 50 000 |
| | 5 | Batch size | 10 |
| | 6 | Discount ($\gamma$) | 0.99 |
| Backprop | 1 | Learning rate | $5 \times 10^{-4}$ |
| | 2 | Adam epsilon | $10^{-5}$ |
| | 3 | Adam $\beta_1$ | 0.9 |
| | 4 | Adam $\beta_2$ | 0.999 |
| | 5 | Buffer size | 50 000 |
| | 6 | Exploration fraction | 0.1 |
| | 7 | Final value of random action probability | 0.02 |
| | 8 | Batch Size | 32 |
| | 9 | Discount ($\gamma$) | 0.99 |
| | 10 | Target update frequency | 500 |
| | 11 | Gradient norm clipping coefficient | 10 |

Table 5: Hyperparameters for Atari domain

| Method | # | Hyperparameter | Value |
|---|---|---|---|
| TAGI | 1 | Horizon | 128 |
| | 2 | Initial standard deviation for the value function ($\sigma_V$) | 2 |
| | 3 | Decay factor ($\eta$) | 0.9999 |
| | 4 | Minimal standard deviation for the value function ($\sigma_V^{\min}$) | 0.3 |
| | 5 | Batch size | 32 |
| | 6 | Discount ($\gamma$) | 0.99 |
| | 7 | Number of actor-learners | 1 |
| Backprop | 1 | Horizon | 5 |
| | 2 | Initial learning rate | $LogUniform(10^{-4}, 10^{-2})$ |
| | 3 | Learning rate schedule | LinearAnneal(1, 0) |
| | 4 | RMSProp decay parameter | 0.99 |
| | 5 | Exploration rate 1 ($\epsilon_1$) | 0.1 |
| | 6 | Exploration rate 2 ($\epsilon_2$) | 0.01 |
| | 7 | Exploration rate 3 ($\epsilon_3$) | 0.5 |
| | 8 | Probability of exploration rate 1 | 0.4 |
| | 9 | Probability of exploration rate 2 | 0.3 |
| | 10 | Probability of exploration rate 3 | 0.3 |
| | 11 | Exploration rate schedule (first four million frames) | Anneal from 1 to $\epsilon_1, \epsilon_2, \epsilon_3$ |
| | 12 | Batch size | 5 |
| | 13 | Discount ($\gamma$) | 0.99 |
| | 14 | Number of actor-learners | 1 |