# OpenReview forum: "Analytically Tractable Bayesian Deep Q-Learning"
_NeurIPS.cc/2021/Conference — NeurIPS 2021 Submitted_

### Official Review · Reviewer_JJGA · 2021-07-15

**Rating:** 5
**Confidence:** 4

**Summary:**

This paper proposes applying TAGI (Goulet et al, 2019), a framework for approximate Bayesian inference in deep neural networks, to deep q-learning. Using assumptions of Gaussianity on various distributions, TAGI infers both the neural network parameters and unit activations, without needing gradient descent. Because all distributions are assumed to be Gaussian, the means and variances can be updated analytically with linear computational complexity. TAGI is essentially used to infer the correct Q-value function, and the approach is applied to lunar lander, cartpole, and several Atari games.

**Limitations And Societal Impact:**

A discussion of societal impact was not relevant for this work, as it was more foundational in nature and would not introduce additional impacts beyond that inherent generically in machine learning.

**Main Review:**

Overall, I liked the idea of introducing Bayesian approaches to deep RL is an interesting, particularly as it offers a more principled way of handling the exploration-exploitation tradeoff compared to epsilon-greedy exploration. However, I do have several reservations on both the theoretical and empirical aspects on this work, and for the reasons given below, I cannot recommend this paper for acceptance as this stage.

Strengths:
* introducing Bayesian inference into deep RL allows for the possibility for a more principled approach to exploration-exploitation via Thompson sampling; exploration is driven by greater posterior uncertainty, while a lower uncertainty will drive greater exploitation
* compared to alternative approaches to Bayesian deep learning, TAGI's inference can be done analytically with linear computational complexity with respect to the number of network parameters

Theoretical weaknesses:
* the authors have chosen not to not use a target network for TAGI-DQN. This is rather controversial as it changes the training target. DQN updates the Q-network towards a fixed TD target. For the learning target to be fixed, it needs a fixed bootstrap, which therefore requires a fixed target network. By not using a fixed target network, TAGI-DQN attempts to infer the network parameters that minimize a different learning objective, what Sutton & Barto (2ed, Ch 11.5) call the mean squared TD error, and this is known to converge to an incorrect result.

Empirical weaknesses:
* the Atari results were ran for 40M frames (only a fifth of the 200M that is more standard). My concern is that this is too early to tell what the asymptotic performance of these algorithms will be like. For instance, it'd be nice to know whether any of TAGI's early leads over the backprop algorithm will be stable over longer horizons. It'd also be nice to know whether TAGI training eventually converges to some asymptotic performance or whether instabilities will emerge over longer training horizons.
* I think this work could also benefit from some ablation studies. For instance, the authors chose not to include a target network for TAGI: as this is a striking design decision, it'd be helpful to see what the effects of it is via an ablation study. Presumably, TAGI explored by Thompson sampling (it was not made explicit in the paper), and it'd be interesting to see how this compares to a more traditional epsilon-greedy policy.
* Finally, it'd be helpful to also examine TAGI in a policy evaluation setting to see whether it can converge onto a correct, known policy.

---Review update---

I thank the authors for their response to my questions during the discussion period. In light of their comments, my previous theoretical concerns have been addressed, and I have raised my score to 5 to reflect that. But I still feel that the paper's baselines were rather thin, and multiple suggestions for improvement have been offered by myself and the other reviewers. I'd like to mention the atari baselines in particular. The authors' current choice is unusual and never used beyond the original paper. The authors compare against n-step Q-learning of Mnih et al (2016), but the 1-thread version of it. This baseline is never used simply because it performs poorly -- it has no replay buffer or parallel actors to make up for it. I appreciate the authors' intention was to demonstrate an improvement in the online learning setting. However, I think it'd still be helpful to compare against a more widely used atari baseline as well, such as DQN and/or the multi-actor version of n-step Q-learning. I also believe it'd be helpful if the authors could provide results on more games, as the current sample size is small -- the authors show TAGI is better in only 3 out of the 5 games they ran. Based on this, it is difficult to conclude whether the RL practitioner should expect TAGI to do better than gradient descent if she were considering training an agent on a new task.

**Time Spent Reviewing:**

3

---

> ### Author Response · Authors · 2021-08-05
> **Rebuttal and Clarifications**
>
> R3C1: The authors have chosen not to not use a target network for TAGI-DQN. This is rather controversial as it changes the training target
>
> Reply: We have chosen not to separate the Q-network with the target network. The objective of using the neural network to approximate the Q function taking as input a state and action does not change. In deterministic deep reinforcement learning, the target network’s parameters are updated with the Q-network’s parameters every T steps in order to ensure the stability during training because the parameter update relies on gradient-based optimization approach, i.e., a point estimate. In addition, this procedure is not suited for online reinforcement learning, i.e., a desired property of reinforcement learning (Kim et al 2019). In this paper, we estimate the posterior of neural networks’ parameters using TAGI rather than a gradient-based approach. The experiments on cart pole and lunar lander environments empirically show that TAGI-DQN outperforms DQN without the need of a target network.
>
> Reference:
> Kim et al 2019: DeepMellow: Removing the Need for a Target Network in Deep Q-Learning, IJCAI-19.
>
> R3C2: The Atari results were ran for 40M frames (only a fifth of the 200M that is more standard)…it'd be nice to know whether any of TAGI's early leads over the backprop algorithm will be stable over longer horizons
>
> Reply: We confirm that the same behaviour is observed with longer horizons of 200M. We did not included these results in the initial submission due to computational limitations.
>
> R3C3: Presumably, TAGI explored by Thompson sampling (it was not made explicit in the paper), and it'd be interesting to see how this compares to a more traditional epsilon-greedy policy
>
> Reply: It was already explicitly mentioned on line119 that TAGI-DQN relies on Thompson sampling. Moreover note that Section 4 already compare the performance of the TAGI-DQN using Thompson sampling, with the traditional DQN using epsilon-greedy policy on the cart pole, lunar lander, and Atari environments, that is, the off-policy and on-policy reinforcement learning.
>
> R4C4: Finally, it'd be helpful to also examine TAGI in a policy evaluation setting to see whether it can converge onto a correct, known policy
>
> Reply: We think that the experiments that we have presented in this paper already demonstrate that TAGI-DQN converges to good policies.

---

> > ### Comment · Reviewer_JJGA · 2021-08-13
> > **Clarification of review comments**
> >
> > I thank the authors very much for taking the time to respond to my comments. I appreciate the further insights their comments have provided. There may have been a few points in my original review which were not entirely clear and which I think I ought to clarify.
> >
> > Comment 1: the authors have chosen not to not use a target network for TAGI-DQN. This is rather controversial as it changes the training target.
> >
> > I apologise for the lack of clarity here. While I do appreciate that TAGI maintains a posterior distribution over weights and unit activations rather than a single point estimate, we know that in the infinite data limit, the posterior will collapse onto a point estimate, and it is this limit that I am concerned about. In gradient-based Q-learning, we know that there are two approaches to computing gradients: semi-gradient methods and full-gradient methods. Both have similar objective functions:
> >
> > $ [ Q(s_t,a_t; \theta) - ( r(s_t) + \gamma  Q(s_{t+1}, a_{t+1}; \theta ) ]^2, $
> > where $a_{t+1}$ is the action taken is the successor state $s_{t+1}$.
> >
> > But they differ in what the gradient operator acts on. In semi-gradient methods, the gradient operator only acts on $Q(s_t,a_t; \theta)$ whereas in full-gradient methods, it also acts on the bootstrap term $Q(s_{t+1}, a_{t+1}; \theta )$. Essentially, semi-gradients regard the bootstrap term as a constant term inside the objective whereas full-gradients do not. Full-gradient methods, however, are known to converge onto solutions that are known to be incorrect, as explained in Sutton & Barto Ch11.5 (2ed). My concern is that in the infinite data limit, TAGI's posterior will collapse on the same learning target as full-gradient methods. Consider TAGI's negative log-likelihood from eq 6:
> >
> > $- \frac{1}{ \sigma_V^2} [ Q(s_t,a_t; \theta) - ( r(s_t) + \gamma  Q(s_{t+1}, a_{t+1}; \theta ) ]^2.$
> >
> > Apart from the factor of $1/ \sigma_V^2$, this is the same as objective above. TAGI attempts to infer parameters $\theta$ that will minimize this log-likelihood, but these parameters affect both $Q(s_t,a_t; \theta)$ as well as $Q(s_{t+1}, a_{t+1}; \theta )$. This results in the same minimization objective as full-gradient methods, except that rather than using gradient descent, TAGI learns these parameters through Bayesian inference. And in the infinite data limit, it will collapse on this solution and not the semi-gradient solution. In comparison, DQN relies on the target network to implement semi-gradient updates, as the gradient operator will not act on the target network. Admittedly, the Kim et al, 2019 paper that the authors mentioned have successfully eliminated the target network, but they still implement semi-gradient updates by hand-coding the gradient operator to not act on the bootstraps.
> >
> > Reply 3: It was already explicitly mentioned on line119 that TAGI-DQN relies on Thompson sampling. Moreover note that Section 4 already compare the performance of the TAGI-DQN using Thompson sampling, with the traditional DQN using epsilon-greedy policy.
> >
> > Ok, I understand now what the authors are talking about in line 119 — I was a bit thrown off when they said “analogously to Thompson sampling”, as it sounds like they were doing something similar but not identical to Thompson sampling. It might be clearer and more direct if the authors simply said that they Thompson sample a set of q-values from the posterior and then select the greedy action from this sample.
> >
> > However, epsilon-greedy policies are also definable within a Bayesian context: when one acts greedily, it would be with respect to the expected value for each action, where the expectation is now taken over the Bayesian posterior as well. Expectations are easy to obtain in the authors' framework as they are already tracking the mean (since they've assumed Gaussian posteriors). Should the authors update the paper, such a comparison would be illuminating to see how much better Thompson sampling is over epsilon-greedy exploration.
> >
> > Comment 4: Finally, it'd be helpful to also examine TAGI in a policy evaluation setting to see whether it can converge onto a correct, known policy.
> >
> > Policy evaluation can offer different insights into the performance of TAGI. TAGI proposes an alternative way of doing neural dynamic programming, but theoretical guarantees on its performance are lacking. So it would be helpful to know how TAGI compares empirically to a known ground truth. Obtaining a ground truth with policy iteration is difficult because of the inherent non-stationarity in the changing policy. Additionally, in the case of Atari, the complexity of the task prevents us from knowing what the true optimal value function should be. Therefore, it’d be helpful to assess TAGI in a setting where the ground truth value function is known. Policy iteration with Atari does not offer as clean of an assessment in this regard. Again, I believe results like these would help validate TAGI as a viable alternative approach to neural dynamic programming and thereby strengthen the conclusions of this paper.

---

> > > ### Author Response · Authors · 2021-08-16
> > > **Rebuttal and Clarifications**
> > >
> > > R3C1: We know that in the infinite data limit, the posterior will collapse onto a point estimate, and it is this limit that I am concerned about…But they differ in what the gradient operator acts on. In semi-gradient methods, the gradient operator only acts on $Q(s_{t}, a_{t};\theta)$  whereas in full-gradient methods, it also acts on the bootstrap term $Q(s_{t+1}, a_{t+1};\theta)$…in the infinite data limit, it will collapse on this solution (full-gradient) and not the semi-gradient solution.
> > >
> > > Reply: We think that the reviewer misunderstood our method that does not separate the Q-network and target network. In TAGI-DQN, the term $Q(s_{t+1}, a_{t+1})$ is a fixed quantity during the TAGI inference step. Therefore, the target value (i.e., learning target) is fixed. According to Mnih et al. (2015), the definition of the target network is that it has the same architecture as the Q-network, yet the target network’s parameters are only updated with the Q-network’s parameters every C steps. The choice of not separating the Q-network with the target network can be seen as C = 1. Therefore, the value of $Q(s_{t+1}, a_{t+1})$ in the target value is obtained directly using the Q-network that acts like a predictor given the input $s_{t+1}$ (see lines 12-15 in Algorithm 1, page 05).
> > >
> > > The limitations of the coexistence of the Q-network and target network:
> > >
> > > (1) It is not well suited for the online reinforcement learning (Kim et al. 2019),
> > >
> > > (2) It introduces lagging in the target value (learning target) because the parameters are not up-to-date (Piché et al. 2021),
> > >
> > > (3) It doubles the memory required to store neural network’s parameter.
> > >
> > > The TAGI-DQN allows addressing those above limitations because of the TAGI’s probabilistic inference capability (See section 2.1, page 02) and Thompson Sampling (See section 2.3, page 03).
> > >
> > > In addition, we agree that the posterior of model parameters will collapse onto a point estimate with the infinite data limit. However, it is not the case in real-world systems that provide limited and noisy data (Dulac-Arnold, Mankowitz, Hester, 2019). For such cases, probabilistic inference is better suited than point estimate approaches.
> > >
> > > References:
> > >
> > > Mnih et al. (2015): Human Level Control Through Deep Reinforcement Learning. Nature.
> > >
> > > Kim et al (2019): DeepMellow: Removing the Need for a Target Network in Deep Q-Learning, IJCAI-19.
> > >
> > > Piché et al. (2021): Beyond Target Networks: Improving Deep Q-learning with Functional Regularization.
> > >
> > > Dulac-Arnold, Mankowitz, Hester, (2019): Challenges of Real-World Reinforcement Learning. ICML.
> > >
> > > R3C2: Consider TAGI's negative log-likelihood from eq 6: $-\tfrac{1}{\sigma_{V}^{2}} [Q(s_{t}, a_{t};\theta) - (r(s_{t}) + \gamma Q(s_{t+1}, a_{t+1};\theta))]^{2}$…TAGI attempts to infer parameters
> > > that will minimize this log-likelihood, but these parameters affect both $Q(s_{t}, a_{t})$ and $Q(s_{t+1}, a_{t+1})$. This results in the same minimization objective as full-gradient methods, except that rather than using gradient descent, TAGI learns these parameters through Bayesian inference.
> > >
> > > Reply: As the reviewer mentioned, we confirm that TAGI does not require an objective function in order to learn the posterior of model parameters because the parameter learning procedure in TAGI is not treated as an optimization problem. Instead, TAGI leverages the Gaussian conditional equations (i.e., Bayesian inference) in order to learn this posterior. An overview of TAGI approach is presented in Section 2.1. More technical details are provided by Goulet et al. (2020).
> > >
> > > References:
> > >
> > > Goulet, Nguyen and Amiri (2020): Tractable approximate Gaussian inference for Bayesian neural networks.
> > >
> > > R3C3: I was a bit thrown off when they said “analogously to Thompson sampling”, as it sounds like they were doing something similar but not identical to Thompson sampling
> > >
> > > Reply: We will correct this ambiguous phrasing.
> > >
> > > R3C4: However, epsilon-greedy policies are also definable within a Bayesian context: when one acts greedily, it would be with respect to the expected value for each action, where the expectation is now taken over the Bayesian posterior as well…Should the authors update the paper, such a comparison would be illuminating to see how much better Thompson sampling is over epsilon-greedy exploration.
> > >
> > > Reply: We think that the additional experiment that the reviewer suggests is not best suited for the main purpose of our paper. The goal of this work is (1) to eliminate the epsilon-greedy policy and (2) to provide a new framework that automatically obtain a policy that balances the exploration and exploitation. TAGI and Thompson sampling work as a team to serve this purpose, where TAGI provides the estimation of the model parameter uncertainty and the Thompson sampling then factors the model parameter uncertainty into the selection of an action. If we employ the epsilon-greedy policy with TAGI, we cannot fully leverage TAGI’s capability on estimating parameter uncertainty.
> > >
> > > Strens (2000) states that a theoretically justifiable approach is
> > >
> > > (1) to retain a notion of uncertainty in the model parameters and,
> > >
> > > (2) to take decisions according to hypotheses for the true model parameters.
> > >
> > > In addition, we have already demonstrated that TAGI-DQN outperformed traditional DQN with the epsilon-greedy policy on cart pole, and lunar lander and Atari (3/5) environments without any additional hyperparameters relating the exploitation-exploration process. The traditional DQN that relies on the epsilon-greedy policy requires tuning the hyperparameters relating to it before training because the epsilon-greedy policy  selects an action using the greedy policy with a probability $1-\epsilon$ and a random action with a probability $\epsilon$.
> > >
> > >  For instance, the use of the epsilon-greedy policy in the traditional DQN results in (1) a hyperparameter (i.e., exploration fraction) for the case of cart pole and lunar lander, and (2) six different hyperparameters (i.e., exploration rate 1, 2 & 3; probability of exploration rate 1, 2 & 3) for Atari games (See Table 4&5 in the supplementary material). This hyperparameter tuning process for the epsilon-greedy policy becomes computationally demanding for real-world applications where we have to deal with a large number of environments across different domains.
> > >
> > > References:
> > >
> > > Strens (2000): A Bayesian Framework for Reinforcement Learning, ICML.
> > >
> > > R3C5: TAGI proposes an alternative way of doing neural dynamic programming, but theoretical guarantees on its performance are lacking. So it would be helpful to know how TAGI compares empirically to a known ground truth.
> > >
> > > Reply: We couldn’t find a known ground truth policy for a complex system having high dimensional action- and state-spaces.
> > >
> > > R3C6: Obtaining a ground truth with policy iteration is difficult because of the inherent non-stationarity in the changing policy. Additionally, in the case of Atari, the complexity of the task prevents us from knowing what the true optimal value function should be.  Therefore, it’d be helpful to assess TAGI in a setting where the ground truth value function is known.
> > >
> > > Reply: The point the reviewer mentions is one of reason why we have benchmarked the performance of TAGI-based approach on different environments (i.e., cart pole, lunar lander, and Atari) with on- and off-policy reinforcement learning algorithms.

---

> > > > ### Comment · Reviewer_JJGA · 2021-08-16
> > > > **Thank you for the clarifications**
> > > >
> > > > I'd like to thank the authors very much for clarifying things. Their latest comments about how the update works have indeed addressed my main theoretical concerns. I am just wondering whether there's any way this detail could be made clearer, especially as the authors do not give the update in line 16 explicitly (which I understand from the need for brevity). One thing I noticed is that the term $\mathbf{y}$ in $PDF(\mathbf{\theta} | \mathbf{y} )$ has not been defined. Although I can now infer from context what the authors mean, one thought might be to define $\mathbf{y}$ explicitly in terms of $r$, $\mathbf{\mu}^y$, $\mathbf{\Sigma}^y$, and $\mathbf{\theta}$ to show more clearly the separation between the Q-network and the bootstrap, as this might help ease understanding for the reader.
> > > >
> > > > At this stage, the only other thing I'd like to request is whether the authors could share the complete Atari results for 200M frames, perhaps through an anonymized link as other authors have done.

---

> > > > > ### Author Response · Authors · 2021-08-17
> > > > > **Clarifications**
> > > > >
> > > > > R3C1: I am just wondering whether there's any way this detail could be made clearer, especially as the authors do not give the update in line 16 explicitly (which I understand from the need for brevity).  One thing I noticed is that the term $\mathbf{y}$ in $PDF(\theta|\mathbf{y})$ has not been defined
> > > > >
> > > > > Reply: As Reviewer 5v2P (a.k.a. Reviewer 2) suggested, the manuscript will be revised to add more technical details related to TAGI’s theory to Section 2.1 in the manuscript. This should address the reviewer’s concerns.
> > > > >
> > > > > R3C2: …define $\mathbf{y}$ explicitly in terms of $r, \mu^{y}, \mathbf{\Sigma}^{y}$, and $\theta$ to show more clearly the separation between the Q-network and the bootstrap, as this might help ease understanding for the reader.
> > > > >
> > > > > Reply: The section 2.2 in the manuscript will be revised in order to improve the clarity regarding the Q-network and the bootstrap.
> > > > >
> > > > > R3C2: At this stage, the only other thing I'd like to request is whether the authors could share the complete Atari results for 200M frames, perhaps through an anonymized link as other authors have done.
> > > > >
> > > > > Reply: Please, find below the anonymized link contains the additional results that the reviewer requested.
> > > > >
> > > > > https://ufile.io/5a7r9ipb
> > > > >
> > > > > We confirm that this link does not contain any identifying information violate the double-blind reviewing policy.

---

> > > > > > ### Comment · Reviewer_JJGA · 2021-08-19
> > > > > > **Thank you for the complete Atari results**
> > > > > >
> > > > > > I'd like to thank the authors for providing the complete Atari results. The authors have correctly replicated the published results in Fig 3 of Mnih et al, 2016, and they have shown that TAGI outperforms the backprop results for 3 out of the 5 games.
> > > > > >
> > > > > > I was wondering if the authors had any insights into the two failure cases -- why might TAGI perform worse in some cases, and how should an RL practitioner know when to use TAGI and when backprop might be better? I'm also wondering whether a larger sample size (i.e. more games) might strengthen the case that using TAGI is on average better than using backprop.
> > > > > >
> > > > > > My other question is whether the authors have also considered comparing TAGI against a vanilla DQN baseline? I appreciate the authors' intentions were to demonstrate TAGI's benefit for online RL algorithms, which DQN is not. But providing DQN as an additional result has the benefit that it is a stronger Atari baseline and one more commonly used by the community, meaning the readership would be more familiar with it. This would also facilitate comparison against other Atari results in the literature.

---

> > > > > > > ### Author Response · Authors · 2021-08-19
> > > > > > > **Rebuttal and clarifications**
> > > > > > >
> > > > > > > R3C1: I was wondering if the authors had any insights into the two failure cases -- why might TAGI perform worse in some cases, and how should an RL practitioner know when to use TAGI and when backprop might be better?
> > > > > > >
> > > > > > > Reply: We have not spent time on identifying why a method performs better than another in some cases because
> > > > > > >
> > > > > > > (1) our ressources are spread thin across an array of TAGI developments & applications,
> > > > > > >
> > > > > > > (2) we do not think that the algorithm presented here is a finality, but only a starting point.
> > > > > > >
> > > > > > > The main goal of this paper is to demonstrate that we are able to leverage the analytically tractable inference engine (i.e., TAGI) on the reinforcement learning (RL) problem and that it can be scale to complex problems such as the Atari game environment, which no other Bayesian neural network has done before. The proposed framework allows addressing two major practical issues (Irpan 2018; Henderson et al. 2018):
> > > > > > >
> > > > > > > (1) automatic balances between the exploration and exploitation
> > > > > > >
> > > > > > > (2) number of hyperparameters to be tuned. For instance, the proposed framework (i.e., TAGI-DQN) reduced approximately half the number of hyperparameters compared to traditional DQN (DQN with experience replay and n-step return) (see Table 4&5 in the supplementary material)
> > > > > > >
> > > > > > > If RL practitioners have faced these issues on their RL applications, TAGI-DQN is a better candidate than traditional DQN (i.e., DQN with experience replay and n-step returns).
> > > > > > >
> > > > > > > We believe the possibilities with TAGI go way beyond the first vanilla implementation presented in this paper and even way beyond the Atari environment; TAGI has the potential to replace backpropagation across the spectrum of network architectures. Although this claim remains to be fully validated, the comparison provided in this paper on complex RL environment is a key step in that direction.
> > > > > > >
> > > > > > > References:
> > > > > > >
> > > > > > > Irpan 2018: Deep Reinforcement Learning Doesn't Work Yet.
> > > > > > >
> > > > > > > Henderson et al. 2018: Deep Reinforcement Learning that Matters
> > > > > > >
> > > > > > > R3C2: I'm also wondering whether a larger sample size (i.e. more games) might strengthen the case that using TAGI is on average better than using backprop.
> > > > > > >
> > > > > > > Reply: We have not benchmarked the performance of TAGI-DQN on a larger sample size because
> > > > > > >
> > > > > > > (1) we have limited computational resources,
> > > > > > >
> > > > > > > (2) as mentioned on line 178-179 in the paper, the five games (i.e., Beamrider, Breakout, Pong, Qbert, Space Invaders)  are commonly selected for tuning the hyperparameters for the entire Atari games (see Mnih et al. 2016; Mnih et al. 2013), meaning that they represent the common features for the other games.
> > > > > > >
> > > > > > > References:
> > > > > > >
> > > > > > > Mnih et al. 2016: Asynchronous methods for deep reinforcement learning. ICML
> > > > > > >
> > > > > > > Mnih et al. 2013: Playing atari with deep reinforcement learning.
> > > > > > >
> > > > > > >
> > > > > > > R3C3: My other question is whether the authors have also considered comparing TAGI against a vanilla DQN baseline? I appreciate the authors' intentions were to demonstrate TAGI's benefit for online RL algorithms, which DQN is not. But providing DQN as an additional result has the benefit that it is a stronger Atari baseline and one more commonly used by the community, meaning the readership would be more familiar with it. This would also facilitate comparison against other Atari results in the literature.
> > > > > > >
> > > > > > > Reply: For the same reason mentioned while replying to R3C1, we have not considered comparing TAGI against a vanilla DQN baseline.

---

### Official Review · Reviewer_5v2P · 2021-07-16

**Rating:** 5
**Confidence:** 4

**Summary:**

This paper adapts a recently proposed analytical closed-form Bayesian approach to optimizing Neural Networks to Q-Learning. The proposed TAGI-DQN architecture is optimized without gradient descent. RL policies based on TAGI-DQN are shown to perform well on a collection of OpenAI Gym environments, including a small number of Atari games.

**Limitations And Societal Impact:**

The authors did an adequate job discussing the computational limitations of their approach as well as the types of contemporary RL methods the TAGI optimization approach could be currently applied to. There are however remaining questions regarding some of the technical limitations of the TAGI framework in light of overfitting, or getting stuck in local optima as the solution method still relies on collecting data proactively while the policy is trained.

**Main Review:**

#### **High-level evaluation**
I am excited about the potential advances to RL in complex environments that may be ushered in by the possible contributions put forward in this paper. I however do not feel that the work, as currently presented, is wholly ready for publication and perhaps another round of revisions (including the addition of more relevant baselines -- see "Relevance" section below) would do this paper a great service. While reading, I got the sense that the authors didn't want to distract from their intended contributions by dwelling on how TAGI works. This is always a tricky balance in writing papers (how much background is too much?) but I did feel that there were too significant of conceptual+technical gaps introduced in the paper by keeping the discussion too high-level and informal through Section 2.

#### **Originality**
The adaptation of the TAGI framework to RL domains is very intriguing. Analytical solutions to a Bayesian treatment of NNs could greatly improve challenges of sample efficiency within RL. This advantage is demonstrated in the experimental analysis across a variety of domains.

#### **Quality and Clarity**
The major weaknesses of this paper concern the clarity of the formulation and utlization of the TAGI with DQNs. In particular, I found the discussion in Section 2.1 to be far too high-level for introducing a relatively complex new approach. A phrase that ran through my head while reading was “show don’t just tell”. The paper is quite light on technical details and it wasn’t due to space constraints as there was nearly 2 pages of unused space remaining in the submitted version of the paper. I feel that a more formal approach to how the two steps of TAGI interact with individual parameter distributions and their connections through hidden layers is necessary and would greatly improve the paper. This would make the discussion in Sections 2.2-2.3 much easier to follow and understand. This is especially true in the discussion starting on line 124 when inference through the TAGI-DQN is described.

Particularly egregious in this lack of clarity is the unmotivated and sudden jump at the end of Section 2.1 to RL. It’s not apparent how the discussion around optimizing parameters via TAGI with a single example or a mini-batch directly relates to learning sequentially with episodes of observations and so on. Additionally, without any formal framing, there’s an extensive burden placed on the reader to know exactly what parts of the following sections correspond to the steps within TAGI. With the connection seemingly drawn to the expected value of selecting an action $a_t$ when conditioned on some state $s_t$, it is also somewhat surprising that the family of Distributional RL (Bellemare, et al; 2017 ICML) approaches went uncited. I acknowledge that Dist. RL does not attempt to pass off as a Bayesian approach but the core component of these approaches is in the implicit distribution formed around the expected value of rewards conditioned on action selection.

In line 100, was the word “discrete” intended in the place of “categorical”?

It’s not clear why n-step returns are preferred? Typically this introduces significant variance in DQN based approaches. Is there something particular about TAGI that necessitates the use of multi-step returns?

The elimination of hyperparameters by moving to TAGI-DQN, described on line 130 seems to only consider those associated with SGD optimization (learning rate, optimization hyperparams, minibatch size, etc…). While this may require several components to tune, the architecture construction and size of replay buffer can have an outsized impact as well. While it’s nice to potentially not worry about hyperparameters associated with SGD, there are still significant choices made in the design of a model.

What’s the rationale for decaying the standard deviation $\sigma_{V}$? Is it to account for drifting biases as the parameters of the model are learned? Is this a “hack” that assumes improved model confidence over time? Is this perhaps too optimistic? What happens if an observed state in a later episode falls out of the assumed distribution? Is TAGI-DQN robust to outliers? These are important questions with regards to avoiding the Q-function approximation overfitting too early.

Again, it’s a shame that so little technical content is provided in Section 2. The primary contribution of this paper is the extension of TAGI to RL domains and architectures. Without the details of how TAGI actually works, line 16 in Algorithm 1 becomes an intellectual dead-end within this paper as written.

In the OpenAI Gym experiments, the initialization of $\sigma_V$ is not mentioned. Additionally, it is not mentioned whether the comparison with the OpenAI default DQN is especially fair or not. For instance, what is the architecture of the DQN model? Is it the same as the TAGI network? Does TAGI use the same amount of computation or more in it’s optimization steps? Are the batch sizes consistent between the two implementations?

For the experiments on Atari, it’s unclear whether n is set to 128 or whether that horizon is for the decay of the standard deviation. While it’s impressive that TAGI outperforms DQN on several of these Atari games, the discussion or analysis about why it fails on some is missing. It would be nice to be able to develop some insight into what types of domains TAGI may not be appropriate for. Are there some complexities in the state spaces of Beam Rider or Space Invaders that make it harder for the solution approach? Do the representations learned by the CNN encounter (or otherwise get stuck in) local minima? I find it interesting that in both of these games, there is a point after which the baseline DQN solution diverges from the TAGI-DQN solution in performance, indicating that it’s learned something that wasn’t accessible to the TAGI network.


#### **Significance**
If properly evaluated the contributions promised through a Bayesian method optimized via analytical means are potentially of great significance. However, I found the technical discussion and experimental demonstration to be mildly unconvincing (partially due to a lack of clarity as discussed above). Clearly, the performance of the TAGI-DQN is demonstrated to be better than a standard DQN model. However, the lack of comparison to comparable approximate Bayesian RL methods or appropriate ablations is disappointing. Without comparison to relevant baselines, it’s hard to feel confident about the proposed analytical solution from a practical standpoint. It’s possibly assumed that TAGI-DQN is automatically more efficient/effective due to the closed-form solution. I say this as a means to rigorously understand the specific advantages borne by the TAGI framework applied to a Bayesian form of the DQN.

In line 144, a “few” BNN approaches that have been applied to large RL domains are alluded to. It would be preferable if these papers are explicitly cited and discussed so as to provide greater context into how TAGI differs (aside from only being an analytical closed-form solution method, if there is anything else?). Is the way the network parameter distributions are factorized to make inference tractable in a similar way to any of these BNN approaches, what about variational inference types of solution methods (such as VariBad--Zintgraf, et al; 2020 ICLR)? How similar is that to TAGI? Answers to these types of questions that would greatly improve the framing of the results that are shown in the following sections of the paper.


##### **Additional references**
Bellemare, Marc G., Will Dabney, and Rémi Munos. "A distributional perspective on reinforcement learning." International Conference on Machine Learning. PMLR, 2017.

Zintgraf, L., et al. "VariBAD: a very good method for Bayes-adaptive deep RL via meta-learning." Proceedings of ICLR 2020 (2020).

**Time Spent Reviewing:**

14

---

> ### Author Response · Authors · 2021-08-05
> **Rebuttal and Clarifications**
>
> R2C1: I did feel that there were too significant of conceptual+technical gaps introduced in the paper by keeping the discussion too high-level and informal through Section 2…Again, it’s a shame that so little technical content is provided in Section 2
>
> Reply: TAGI relies on a totally different approach than gradient-based optimization. For a complete presentation of the method, its benchmark against other methods and a discussion of its limitations, the reader should refer to the paper by Goulet, Nguyen and Amiri (2020)
>
> References:
> Goulet, Nguyen and Amiri (2020): Tractable approximate Gaussian inference for Bayesian neural networks.
>
> R2C2: In line 100, was the word “discrete” intended in the place of “categorical”?
>
> Reply: Yes, it is.
>
> R2C3: It’s not clear why n-step returns are preferred? Typically this introduces significant variance in DQN based approaches. Is there something particular about TAGI that necessitates the use of multi-step returns?
>
> Reply: n-step returns are chosen because (1) it is a faster algorithm than DQN with experiment replay when training on Atari games, (2) we employ an on-policy reinforcement learning method that is well suited for TAGI because of its probabilistic learning capability, and (3) the n-step return is well-established in the literature, therefore, it is convenient for comparison purpose.
>
> There is nothing in particular about TAGI that requires the use of multi-step returns. For the Atari game, n-step returns, Advanced actor critic (A2C) (Mnih et al., 2016), Proximal Policy Optimization (PPO) (Schulman et al., 2017) trained using backpropagation have also used the multi-step returns of 5, 5 and 128 steps, respectively.
>
> References:
> Mnih et al., 2016.        : Asynchronous Methods for Deep Reinforcement Learning,
> Schulman et al., 2017 : Proximal Policy Optimization Algorithms
>
> R2C4: The elimination of hyperparameters by moving to TAGI-DQN, described on line 130 seems to only consider those associated with SGD optimization… While it’s nice to potentially not worry about hyperparameters associated with SGD, there are still significant choices made in the design of a model.
>
> Reply: Yes, the tuning process is still required for the network architecture.
>
> R2C5: What’s the rationale for decaying the standard deviation \sigma_V?
>
> Reply: This approach is similar to what is done in standard deep neural networks trained with backpropagation where noise is added to the gradient (Neelakantan et al., 2015) with a decay schedule.
>
> References:
> Neelakantan et al., 2015: Adding gradient noise improves learning for very deep networks
>
> R2C6: In the OpenAI Gym experiments, the initialization of \sigma_V is not mentioned
>
> Reply: All experiments using TAGI-based approach employ the same initialization procedure for \sigma_V where \sigma_V is initialized at 2 and it is decayed each 128 steps with a factor of 0.9999. This procedure has already been mentioned on lines 198-199 as well as in the supplementary material (Table 4&5).
>
> R2C7: For instance, what is the architecture of the DQN model?
>
> Reply: Both TAGI-DQN and DQN trained with backpropagation use the same network architecture across all experiments. The network architectures are mentioned on line 181-183 for the off-policy RL experiment and on lines 189-191 for the on-policy RL experiment. The further details of network architectures are already provided in Section 2 in supplementary material.
>
> R2C8: Does TAGI use the same amount of computation or more in it’s optimization steps?
>
> Reply:  The training speed of TAGI for the experiment of the off-policy deep RL is approximately three times slower on CPU calculations than the backpropagation-trained counterpart. The reason behind this slower training time is because of its intrinsically different inference engine, so that TAGI’s implementation is not compatible with existing libraries such as TensorFlow or Pytorch. TAGI’s library development is still ongoing and it is not yet fully optimized for computational efficiency. The discussion regarding computational time has already been mentioned on lines 217-222.
>
> R2C9: Are the batch sizes consistent between the two implementations?
>
> Reply: For the cart pole and lunar lander environments, the batch size of 32 for the DQN trained with backpropagation and is kept the same as the openAI’s implementation, while TAGI uses a batch size of 10. For the Atari games, n-step return trained with backpropagation uses a batch size of 5 while TAGI uses a batch size 32. The details for the hyperparameters were already provided in supplementary material (Table 4&5).
>
> R2C10: For the experiments on Atari, it’s unclear whether n is set to 128 or whether that horizon is for the decay of the standard deviation
>
> Reply: For Atari games, both the number of step returns n and the decaying step is set to 128. The details for the hyperparameters were already provided in supplementary material (Table 5).
>
> R2C11: Are there some complexities in the state spaces of Beam Rider or Space Invaders that make it harder for the solution approach?
>
> Reply: It could be.
>
> R2C12: Do the representations learned by the CNN encounter (or otherwise get stuck in) local minima?
>
> Reply: Yes, local optima are a well-known issue in deep RL (Arpan, 2018)
>
> References:
> Arpan, 2018: Deep reinforcement learning doesn’t work yet
>
> R2C13: I find it interesting that in both of these games, there is a point after which the baseline DQN solution diverges from the TAGI-DQN solution in performance, indicating that it’s learned something that wasn’t accessible to the TAGI network
>
> Reply: It could be, but we could say the same for other games such as Breakout, Pong,  and Q-bert where TAGI learned something that the DQN trained with backpropagation cannot.
>
> R2C14: However, the lack of comparison to comparable approximate Bayesian RL methods or appropriate ablations is disappointing
>
> Reply: As mentioned in Section 3, although some Bayesian methods have shown to be capable of tackling classification tasks on datasets such ImageNet, none of them have been applied on large-scale RL benchmark problem such as the Atari environement.
>
> R2C15: It’s possibly assumed that TAGI-DQN is automatically more efficient/effective due to the closed-form solution. I say this as a means to rigorously understand the specific advantages borne by the TAGI framework applied to a Bayesian form of the DQN.
>
> Reply: Yes.
>
> R2C16: In line 144, a “few” BNN approaches that have been applied to large RL domains are alluded to. It would be preferable if these papers are explicitly cited and discussed so as to provide greater context into how TAGI differs (aside from only being an analytical closed-form solution method, if there is anything else?)
>
> Reply: We should have use “no” rather than “few” as we could not find any in the literature.
>
> R2C17: Is the way the network parameter distributions are factorized to make inference tractable in a similar way to any of these BNN approaches, what about variational inference types of solution methods (such as VariBad--Zintgraf, et al; 2020 ICLR)? How similar is that to TAGI?
>
> Reply: Like most of the variational approaches, TAGI uses a diagonal structure for the covariance of posterior. However, the main difference relies on the inference engine that is analytical for TAGI and backpropagation-based for variational approaches.
>
> R2C18: There are however remaining questions regarding some of the technical limitations of the TAGI framework in light of overfitting, or getting stuck in local optima as the solution method still relies on collecting data proactively while the policy is trained.
>
> Reply: TAGI could be stuck in local optimal. An empirical study on the overfitting issue for a toy regression problem conducted by Goulet, Nguyen and Amiri (2020) shows that TAGI does not immune to the overfitting
>
> References:
> Goulet, Nguyen and Amiri (2020): Tractable approximate Gaussian inference for Bayesian neural networks.

---

> > ### Comment · Reviewer_5v2P · 2021-08-09
> > **Thank you for the clarifications and other responses**
> >
> > Thanks for providing some clarification as well as taking the time to answer several of my questions. I appreciate getting a fuller perspective of the submitted paper and will take this with me into the discussion period with the other reviewers.
> >
> > While I do not have my thoughts wholly collected in relation to the responses provided, I would like to suggest again that future revisions of this paper (including the event of it possibly being accepted to this conference) do include a detailed walkthrough of the technical foundations of TAGI. I acknowledge that it feels a bit silly to re-hash what is written in prior work, but when introducing concepts that depart from standards in one community (as the authors have clearly articulated here) it is exceedingly helpful to place those technical foundations in formal context with the writing in the paper. This makes the connections easier to follow as well as better appreciated.
> >
> > As a final high-level recommendation here. I would presume that the paper would be much improved if several of the type of clarifications provided here were also included in the paper. One particular instance of this is the motivation for the decay of the $\sigma_V$ term. Without connecting this to the cited reference, it felt like an ad hoc design decision that accelerated convergence. To touch on this point a little further... Despite this being a previous standard in the training of deep neural networks with SGD, I'm not totally convinced that it's a one-to-one analogue with the TAGI-DQN training paradigm. Without providing a more formal distinction for why this matters (and maybe investigating some of the consequences -- such as overconstraining the parameter distribution too early in the learning process as was pointed out in my review) the choice does still a little ad hoc. The same line of concern remains with the choices of n-step training, etc.
> >
> > By the way, was the OpenAI baseline trained with n-step training? Just needing to be sure that the comparison is totally fair... Based a quick re-read of Section 4 in the submitted paper, it is not totally clear.

---

> > > ### Author Response · Authors · 2021-08-09
> > > **Rebuttal and Clarifications**
> > >
> > > R2C1: Despite this being a previous standard in the training of deep neural networks with SGD, I'm not totally convinced that it's a one-to-one analogue with the TAGI-DQN training paradigm… Without providing a more formal distinction for why this matters (and maybe investigating some of the consequences — such as overconstraining the parameter distribution too early in the learning process as was pointed out in my review) the choice does still a little ad hoc
> > >
> > > Reply: In the case of gradient-based learning, this noise consists in discrete samples added to the gradient itself whereas for TAGI, it consists in additional variance on the output layer so that the update during the inference step will put more weight on the prior rather than on the likelihood, with this effect diminishing with time (Nguyen et Goulet, 2021).
> > >
> > > We agree that more investigations on these hyperparameters need to be conducted, yet we showed in this paper that all hyperparameters related to \sigma_V can be kept constant across completely different environments such as cart pole, lunar lander, and Atari games. Moreover, since these hyperparameters were not subject to grid-search in order to optimize the performance, the results obtained here are representative of what a user should obtain by simply adapting the hyperparameters to fit the specificities and scale of the environment at hand.
> > >
> > > Note here that the DQN trained with backpropagation requires defining hyperparameters such as the schedule for the learning rate and the exploitation rate (see Table 5 in the supplementary material), yet there is no clear explanation where they comes from.
> > >
> > > Reference:
> > >
> > > Nguyen et Goulet, 2021. Analytically Tractable Inference in Deep Neural Networks.
> > >
> > > R2C2: The same line of concern remains with the choices of n-step training.
> > >
> > > Reply: As mentioned in the previous reply. There is nothing in particular about TAGI that requires the use of multi-step returns. The reasons for this choice are
> > >
> > > (1) It is a faster algorithm than DQN with experience replay when training on Atari games, so we were able to run this benchmark with our limited computing resources.
> > >
> > > (2) We explicitly wanted to showcase the capacity of TAGI on on-policy reinforcement learning where the agent is required to modify the policy according to the changes in the environment and only sees the data once (Dulac-Arnold, Mankowitz, Hester, 2019).
> > >
> > > (3) the n-step return is well-established in the literature (Mnih et al., 2016.), therefore, it is convenient for comparison purpose.
> > >
> > > References:
> > >
> > > Dulac-Arnold, Mankowitz, Hester, 2019, Challenges of Real-World Reinforcement Learning.
> > >
> > > Mnih et al., 2016.  Asynchronous Methods for Deep Reinforcement Learning.
> > >
> > > R2C3: By the way, was the OpenAI baseline trained with n-step training? Just needing to be sure that the comparison is totally fair.
> > >
> > > Reply: The results are those provided by Minh et al. (2016) which were trained using n-step return algorithm. It was already mentioned on line 210.
> > >
> > > References:
> > >
> > > Mnih et al., 2016.  Asynchronous Methods for Deep Reinforcement Learning.

---

> > > > ### Comment · Reviewer_5v2P · 2021-08-25
> > > > **Apologies for the delay**
> > > >
> > > > Apologies for the delay in my response. I've enjoyed the authors explaining and clarifying things for me. It has greatly helped in the discussion between reviewers.
> > > >
> > > > My major recommendation for future revisions of this work is that many of the questions I have posed would be easy to avoid by being more explicit in the paper. Most readers of the submitted work will not know every detail of the methods used to build and benchmark the TAGI-DQN. It is best practice to provide sufficient minimal detail so things are absolutely clear. A stray citation usually does not suffice for this purpose.
> > > >
> > > > Additionally, more thorough comparisons with contemporary approximate Bayesian approaches to RL are warranted. Demonstrating the performance of TAGI-DQN against these and further showing the latent advantages of the proposed analytic approach in some way would go a long way toward improving this paper.

---

> > > > > ### Author Response · Authors · 2021-08-26
> > > > > **Clarifications**
> > > > >
> > > > > R2C1: My major recommendation for future revisions of this work is that many of the questions I have posed would be easy to avoid by being more explicit in the paper. Most readers of the submitted work will not know every detail of the methods used to build and benchmark the TAGI-DQN.
> > > > >
> > > > > Reply: The manuscript will be revised for addressing the reviewer’s questions and for improving the clarity. More technical details related to the method, i.e., TAGI that we employ in this paper will be added to the manuscript.
> > > > >
> > > > > R2C2: Additionally, more thorough comparisons with contemporary approximate Bayesian approaches to RL are warranted. Demonstrating the performance of TAGI-DQN against these and further showing the latent advantages of the proposed analytic approach in some way would go a long way toward improving this paper.
> > > > >
> > > > > Reply: In the literature, we have not found any Bayesian deep learning methods such as variational inference and sampling methods (e.g., Hamiltonian Monte Carlo) which estimate the posterior for the network’s parameters & then use the information from this posterior for the selection of an action given a state, and which was scalable enough to be applied to the Atari games’ environment.

---

### Official Review · Reviewer_tBCT · 2021-07-17

**Rating:** 3
**Confidence:** 3

**Summary:**

The paper proposes to use TAGI, an approach for training BNNs, within DQN to enable Thompson sampling.

**Limitations And Societal Impact:**

Yes

**Main Review:**

The paper has very limited novelty,  simply plugging TAGI (a previously proposed approach for training BNNs) into DQN. The results are also not particularly impressive, being run on a very limited set of environments and being outperformed by vanilla DQN on several. While having fewer hyperparameters to tune than training the network via gradient descent is a nice benefit, a more thorough evaluation would be needed to demonstrate the advantages of TAGI in reinforcement learning.

Why not compare to Thompson Sampling with Bayesian approaches to Q learning like in Azizzadenesheli or Osband et al? This seems like the more direct comparison than vanilla DQN.

The authors also describe their learning algorithm as being analytic, but still relies on repeated iterations, so it's unclear to me what the benefit of the "analytic" updates are. If the key benefit here is just the uncertainty is analytically computed when evaluating the network on new inputs, then the authors should compare the evaluation times of TAGI vs other approximate BNN methods that require multiple forward passes.

**Time Spent Reviewing:**

2

---

> ### Author Response · Authors · 2021-08-05
> **Rebuttal and Clarifications**
>
> R1C1: The paper has very limited novelty, simply plugging TAGI (a previously proposed approach for training BNNs) into DQN. The results are also not particularly impressive, being run on a very limited set of environments and being outperformed by vanilla DQN on several
>
> Reply: The objective of our paper is to offer the AI/ML community a new perspective on probabilistic inference for Bayesian deep reinforcement learning which, up to now has been relying on gradient-based methods. The experiments in this paper show that TAGI scales to larger problems and we could reach a performance comparable backpropagation-trained networks while using an analytical inference method requiring fewer hyperparameters.
>
> R1C2: Why not compare to Thompson Sampling with Bayesian approaches to Q learning like in Azizzadenesheli or Osband et al? This seems like the more direct comparison than vanilla DQN
>
> Reply: A Bayesian deep learning approach quantifies the posterior for the parameters of the network. The paper by Azizzadenesheli is not truly a Bayesian neural network as it only considers Bayesian linear regression on the output layer of the network, similarly, Osband et al is a bootstrapped DQN approach and not a Bayesian one. In the literature, we have not found any Bayesian deep learning approach which estimate the posterior for the network’s parameters and which was scalable enough to be applied to the Atari games’ environment.
>
> R1C3: The authors also describe their learning algorithm as being analytic, but still relies on repeated iterations, so it's unclear to me what the benefit of the "analytic" updates are
>
> Reply: the “learning algorithm as being analytic” means that the posterior for neural network’s parameters i.e. p(\theta | data) is inferred using a closed-form analytical method i.e. TAGI. For decades, the application of analytic Bayesian inference on this posterior for a large neural networks has been considered to be intractable (Goodfellow, Bengio, and Courville, 2016).
>
> References:
> Goodfellow, Bengio, and Courville, 2016: Deep Learning, MIT press

---

> > ### Comment · Reviewer_tBCT · 2021-08-17
> > **Response to Author Rebuttal and Clarifications**
> >
> > Thanks for the response.
> >
> > Unfortunately, I don't see the inherent value of having a method that performs (approximate) analytic Bayesian updates, and so believe comparisons to other approximate Bayesian techniques are necessary to properly evaluate the method. I also do not believe the methods of Azzizzadenesheli and Osband can be disregarded simply because they are not "truly Bayesian".

---

> > > ### Author Response · Authors · 2021-08-18
> > > **Rebuttal and Clarifications**
> > >
> > > R1C1: Unfortunately, I don't see the inherent value of having a method that performs (approximate) analytic Bayesian updates
> > >
> > > Reply: Bayesian neural networks i.e., Bayesian deep learning (MacKay, 1992; Neal, 1995; Gal, 2016) aims to obtain the posterior distribution of neural network’s parameters $\theta$ conditional on the training data available $\mathcal{D}$, i.e., $p(\theta|\mathcal{D})$. This posterior is then used to evaluate the predictive posterior of an outcome  of the neural network $\mathbf{y}$ on unseen input data $\mathbf{x}$, i.e., $p(\mathbf{y}|\mathbf{x},\theta)$ . The posterior of neural network’s parameters has been considered to be intractable (no closed-form formulation exists) (Goodfellow, Bengio, and Courville, 2016). Many attempts have been made to address this intractability. Most of modern approaches for Bayesian neural networks can be categorized in two main groups: sampling method and variational inference.
> > >
> > > The sampling method consists in using carefully engineered random samples to approximate the intractable distribution. The most common approches include Markov Chain Monte Carlo and Hamiltonian Monte Carlo (Neal et al. 2011). These methods are guaranteed to converge to the true distribution, yet they are extremely computationally demanding when applied to deep neural networks (Izmailov et al. 2021).
> > >
> > > Variational inference approaches (Kingma, Salimans, and Welling, 2015; Blundell et al., 2015) aim to approximate the posterior of neural network’s parameters with a variational distribution having a known functional forms whose the variational parameters are obtained via the optimization of a loss function. The main drawback is that the expected values of the log-likehood function in the variational inference loss function is commonly intractable (Goan and Fookes, 2020). Monte Carlo integration is commonly used to approximate these expected values, yet it introduces a large variance on the inference process.
> > >
> > > TAGI (Goulet, Nguyen, and Amiri, 2020) leverages Gaussian conditional equations to analytically infer the posterior of neural network’s parameters without the need of numerical approximations (i.e., sampling techniques or Monte Carlo integration) or optimization, while having the same computational complexity as deterministic neural networks relying on backpropagation.
> > >
> > > References:
> > >
> > > Neal, 1995. Bayesian learning for neural networks. PhD thesis.
> > >
> > > MacKay, 1992: A practical Bayesian framework for backpropagation networks, Neural Computation.
> > >
> > > Gal, 2016: Uncertainty in Deep Learning, PhD thesis (see Section 2.2 and Chapter 3).
> > >
> > > Goodfellow, Bengio, and Courville, 2016: Deep Learning, MIT press
> > >
> > > Kingma, Salimans, and Welling 2015: Variational Dropout and the Local Reparameterization Trick, Neurips.
> > >
> > > Blundell et al. 2015: Weight uncertainty in neural networks, ICML.
> > >
> > > Neal 2011:MCMC using Hamiltonian dynamics.
> > >
> > > Izmailov et al. 2021: What Are Bayesian Neural Network Posteriors Really Like?, ICML.
> > >
> > > Goan and Fookes, 2020: Bayesian neural networks: An introduction and survey, Springer.
> > >
> > >
> > >
> > > R1C2: so believe comparisons to other approximate Bayesian techniques are necessary to properly evaluate the method
> > >
> > > Reply: As we mentioned previously, we have not found in the literature any Bayesian deep learning (BDL) approaches (see the definition of BDL in the reply of R1C1),  which estimate the posterior for the network’s parameters and which are scalable enough to be applied to the Atari games’ environment.
> > >
> > > R1C3: I also do not believe the methods of Azzizzadenesheli and Osband can be disregarded simply because they are not "truly Bayesian"
> > >
> > > Reply: We think the reviewer misunderstood our intention from the previous responses regarding the reviewer’s suggestion on comparing our approach with other Bayesian approaches (i.e., we understood as Bayesian neural networks).  If the reviewer wanted to see the comparison of our approach with Bayesian Neural networks on Deep Q-learning, they are not best suited for this purpose because of the same reasons that we mentioned in the previous responses which are
> > >
> > > (1) the approach proposed by  Azizzadenesheli et al., (2018) is not a Bayesian neural networks (see the reply in R1C1) because it only considers Bayesian linear regression on the output layer of the network, but not the remaining layers,
> > >
> > > (2) the bootstrapped DQN approach (Osband et al., 2016) is not a Bayesian neural network, that is already explicitly mentioned in their paper
> > >
> > > We confirm that we do not disregard these two approaches for any reasons. Instead, we appreciate these approaches that take into account the model uncertainty for efficient exploration. As a matter of fact, we mentioned explicitly these approaches and their improvements over deterministic DQN on lines 159-164 in Section 3.
> > >
> > > References:
> > >
> > > Azizzadenesheli et al., 2018: Efficient exploration through Bayesian deep q-networks, IEEE.
> > >
> > > Osband et al., 2016: Deep exploration via bootstrapped dqn. Neurips

---

### Decision · Program_Chairs · 2021-09-27

**Decision:**

Reject

**Comment:**

The reviewers have raised valid concerns that were not adequately addressed by the rebuttal, placing the paper below the bar.